# Evaluation of Immune Status in Two Cohorts of Atlantic Salmon Raised in Different Aquaculture Systems (Case Study)

**DOI:** 10.3390/genes13050736

**Published:** 2022-04-22

**Authors:** Hege Lund, Anne Bakke, Preben Boysen, Sergey Afanasyev, Alexander Rebl, Farah Manji, Gordon Ritchie, Aleksei Krasnov

**Affiliations:** 1Faculty of Veterinary Medicine, University of Life Sciences (NMBU), 1433 Ås, Norway; hege.lund@nmbu.no (H.L.); anne.flore.bakke@nmbu.no (A.B.); preben.boysen@nmbu.no (P.B.); 2I. M. Sechenov Institute of Evolutionary Physiology and Biochemistry, Torez 44, 194223 Saint Petersburg, Russia; afanserg@mail.ru; 3The Leibniz Institute for Farm Animal Biology (FBN), Wilhelm-Stahl-Allee 2, 18196 Dummerstorf, Germany; rebl@fbn-dummerstorf.de; 4Mowi Norway AS, Sandviksboder 77AB, 5035 Bergen, Norway; farah.manji@mowi.com (F.M.); gordon.ritchie@mowi.no (G.R.); 5Nofima AS, Osloveien 1, 1430 Ås, Norway

**Keywords:** Atlantic salmon, aquaculture systems, immune competence, cohort, gene expression, antibodies

## Abstract

Assessment of immune competence of farmed Atlantic salmon is especially important during smoltification and the first several months in the sea. Recently developed tools were applied to salmon raised in a traditional flow-through facility (FT, cohort 1) and in a recirculation aquaculture system (RAS, cohort 2). Fish were sampled at four time-points: parr, smolt, and at three weeks and three months after seawater transfer (SWT); expression of 85 selected immune and stress genes, IgM transcripts (Ig-seq), and circulating antibodies were analyzed. A steady increase in gene expression was seen over time in gill and spleen in both cohorts, and especially in antiviral and inflammatory genes in the gill. Differences between the cohorts were greatest in the dorsal fin but later leveled off. Comparison with a gill reference dataset found a deviation in only three of 85 fish, suggesting a good immune status in both cohorts. Levels of both specific and nonspecific antibodies were higher in cohort 2 in smolts and in growers three weeks after SWT; however, levels evened out after three months in the sea. Ig-seq indicated association between antibody production, expansion of the largest clonotypes, and massive migration of B cells from spleen to gill in smolts. The results suggested greater agitation and higher reactivity of the immune system in RAS-produced salmon, but the difference between the cohorts leveled off over time.

## 1. Introduction

Atlantic salmon aquaculture is undergoing rapid development, and new technologies are being introduced to address issues of fish health and welfare as well as environmental concerns. Flow-through (FT) aquaculture systems belong to the traditional production systems, whereas recirculation aquaculture systems (RAS), which reduce the consumption of water and prolong the rearing of fish in land-based facilities, are among the main innovations, and their share in smolt production is rapidly increasing [1,2,3]. As the farming environment is changing substantially with unknown consequences for fish, the impact of new technologies on salmon biology is a hot spot in aquaculture research [4,5].

The immune system of Atlantic salmon is influenced by a combination of factors including production protocols, farming conditions, developmental changes, and infection pressure. The occurrence of diseases is greatest after seawater transfer and during the first several months in the sea [6,7]. Higher susceptibility of fish in this period can be associated with stress, adaptation to the marine environment, and immune suppression during smoltification, which may affect both the innate and adaptive arms of the immune system [8,9]. Decreased expression of multiple immune genes in smolts, which persists for several months in the sea, has been shown in various independent studies [8,9]. Differences in the composition of affected genes and magnitude of changes suggest this downregulation as a side effect of massive endocrine regulation and tradeoff between immunity and transformation of osmoregulation. The character and magnitude of these changes vary and are most likely influenced by farm conditions and production protocols.

This case study assessed the immune status of two cohorts of Atlantic salmon produced at a traditional FT facility and a RAS facility within the period from parr to three months after seawater transfer (SWT), using recently developed tools. The multigene expression assay (MGE) is narrowing a gap between qPCR analyses of small sets of immune genes that has proven insufficient for diagnostics and high-throughput methods (“omics”) not suitable for large numbers of samples due to high costs, limited capacity, and complexity of data management. We developed an MGE assay for the assessment of immune competence of Atlantic salmon (ImCom) on the Biomark HD platform. Immune and stress genes were selected from Nofima’s transcriptome database according to the scale and reproducibility of responses to pathogens [10]. ImCom has been tested on a large number of fish from different parts of Norway, and a gill reference data set (GRDS) has been compiled [11]. Comparison between smolts with good, intermediate, and poor performance identified the typical problems: coordinated downregulation (suppression) or unhealthy upregulation of genes due to inflammation or infection, and deviation from homeostasis shown by simultaneously increased or decreased expression of large groups of genes. In addition, two integrative metrics based on the entire assay are especially useful for diagnostics: VRG—virus responsive genes [12] and AIS—markers of acute inflammation and stress (9 and 11 genes in the assay, respectively).

Multiplexed antibody assays allow comparison of responses to various antigens in the same sample. In addition to assessment of specific antibody responses to antigens delivered with vaccines, we included nonspecific antibodies (NSAB). NSAB can be detected in the assay by binding circulating antibodies to model antigens toward which the Atlantic salmon has never been exposed. Our recent studies showed strong responses of NSAB to vaccination, functional disorders and infections with pathogens indicating their importance in Atlantic salmon [13,14,15,16]. Sequencing of the variable region of IgM heavy chain (Ig-seq) evaluates the size and complexity of the antibody repertoire [17]. Each unique sequence (clonotype) serves as a barcode for clonal B cells derived from a single progenitor cell. The sharing of identical clonotypes between tissues reflects a migration of B cells [15].

The reported study assessed these tools on field material: Atlantic salmon reared in commercial aquaculture.

## 2. Materials and Methods

### 2.1. Production Protocols

Two cohorts of Atlantic salmon reared in Southwestern Norway were examined (Figure 1). Fish material was provided by Mowi AS (Bergen, Norway), and fish used were from the Mowi strain. No specific disease traits were selected for, and the only trait selected for was growth. The sex ratio in the cohorts was close to unity. Random sampling of fish was performed, and sex was not taken into account. In cohort 1, fish were raised in a traditional flow-through (FT) production unit until transfer to the sea. Fish were fed a commercial freshwater feed (Skretting, Stavanger, Norway). The production unit had an average water temperature per month varying from 9.48 °C to 14.04 °C, the highest temperature measured during the month of March. A 12:12 L:D photoperiod regime was applied until the time of vaccination, followed by a 24:00 constant light regime for 5 weeks until seawater transfer (SWT). Fish were vaccinated intraperitonially (i.p.) with a commercial multivalent vaccine (AquaVac PD7, MSD AH) at a size of 50–70 g. SWT of fish with an average weight of 87.4 (+/−12.5) g took place at the end of August.

Fish in cohort 2 were kept in a recirculation aquaculture system (RAS) with >95% recirculation for the final 11 weeks before SWT. Oxygen levels were measured daily and maintained at >90%. Fish were fed a commercial freshwater feed (Skretting, Stavanger, Norway). The production unit had an average water temperature of 13.5 °C. A 12:12 photoperiod regime was applied up until vaccination and for 5 weeks after vaccination, followed by 24:00 regime for the final 5 weeks until SWT, which took place at an average weight of 113.85 (+/−24.8). Fish were vaccinated i.p. with commercial vaccines (Alphaject Micro 6 and Alphaject Micro 1 PD, Pharmaq, Olso, Norway) at an average weight of 50.5 g. Fish health and performance and facility production parameters were regularly monitored by on-site staff and fish-health veterinarians.

### 2.2. Sampling Regime and Analysis of Fish

Sampling regimes are included in Figure 1 (n = 15 at each time-point). In cohort 1 (FT production unit), samplings were performed shortly before vaccination and the start of constant light stimulation (1st sampling, July) and before SWT to cage farm at 335 degree-days (DD) after vaccination (2nd sampling, August). At the second sampling, fish condition factor and smolt index were evaluated. Smolt index was assessed using a commercial standard visual scoring system (scale 1–4) that evaluated body silver color (4 = most silvery), parr marks (4 = no parr marks), and darkening of the fins (0 = no darkening). Further, samples were collected 3 weeks after SWT (3rd sampling, September) and approximately 3 months after SWT (4th sampling, December).

In cohort 2 (RAS), samples were collected shortly before vaccination (1st sampling, September) and before SWT at 754 DD after vaccination (2nd sampling, October). Fish condition factor and smolt index were assessed as above. Fish were further sampled at 3 weeks after SWT (3rd sampling, November) and at approximately 3 months after SWT (4th sampling, January).

For lethal sampling, fish were anaesthetized with tricaine (Pharmaq, Oslo, Norway) and killed with a blow to head. Organ samples including gill (two filaments from first left gill arch), dorsal fin (distal part of the first ray to head), and spleen were placed in RNALater (Thermo Fisher Scientific, Waltham, MA, USA) and stored overnight at 4 °C before further storage at −80 °C. Blood was collected from the caudal vein with a vacutainer containing EDTA anticoagulant (VWR), and plasma was immediately separated with centrifugation before further storage at −80 °C.

### 2.3. RNA Isolation

Small pieces of dorsal fin, gill, and spleen (5–10 mg) were placed in tubes with 400 µL lysis buffer (Qiagen, Düsseldorf, Germany) and beads, and 20 µL proteinase K (50 mg/mL) was added in each tube. Samples were homogenized in FastPrep 96 (MP Biomedicals, Eschwege, Germany) for 120 s at maximum shaking, centrifuged, and incubated at 37 °C for 30 min. RNA was extracted on Biomek 4000 robot using Agencourt RNAdvance Tissue kit according to the manufacturer’s instructions. RNA concentration was measured with NanoDrop One (Thermo Fisher Scientific, Waltham, MA, USA), and quality was assessed with Agilent Bioanalyzer 2100.

### 2.4. Sequencing of the Variable Region of IgM Heavy Chain

Analyses were performed in spleen and gill as described in [15]. Synthesis of cDNA was primed with oligonucleotide to the constant region (CH) of Atlantic salmon IgM (TAAAGAGACGGGTGCTGCAG), using SuperScript IV reverse transcriptase (Thermo Fisher Scientific, Waltham, MA, USA) according to the manufacturer’s instructions. Libraries were prepared with two PCR reactions. The first PCR amplified cDNA with a degenerate primer, TCGTCGGCAGCGTCAGATGTGTATAAGAGACAGTGARGACWCWGCWGTGTATTAYTGTG, which aligns to the 3′-end of all Atlantic salmon VH genes and a primer GTCTCGTGGGCTCGGAGATGTGTATAAGAGACAGGGAACAAAGTCGGAGCAGTTGATGA to the 5′-end of CH. Both primers were complementary to Illumina Nextera adaptors. Reaction mixtures (20 µL) included 10 µL 2X Platinum™ Hot Start PCR Master Mix (Thermo Fisher Scientific, Waltham, MA, USA), 0.5 µL of each primer (10 pmol/µL), 8 µL water, and 1 µL template. The second PCR used Nextera™ XT Index Kit v2 (Illumina, San Diego, CA, USA), and reaction included 2 µL each primer and 2 µL product of first PCR. PCR program included heating: 1 min at 94 °C, amplification: 10 s at 94 °C, 20 s at 53 °C, and 20 s at 72 °C (30 cycles in first PCR and 9 cycles in second PCR) and extension: 5 min at 72 °C. DNA concentration was measured with Qubit (Thermo Fisher Scientific, Waltham, MA, USA). Aliquots of libraries were combined and purified twice with Qiagen PCR clean-up kit. Sequencing was carried out using Illumina MiSeq™ Reagent Kit v3, 150-cycle (Illumina, San Diego, CA, USA). Libraries were diluted to 4 nM and PhiX control was added to 0.8 nM. After the trimming of Illumina adaptors and primers and removal of low-quality reads, sequences were transferred in a relational database. The frequency of each unique sequence (clonotype) and cumulative frequencies of fifty largest clonotypes (CF) were calculated. Traffic of B cells between the spleen and gill was assessed by the co-occurrence of the largest clonotypes: the hundred largest clonotypes of the spleen (gill) detected also in the gill (spleen) at frequency > 10^−4^.

### 2.5. Multigene Expression Assays

The selection of genes, design, and validation of the assay are described in [11,16]. Several genes that have not shown high diagnostic value were substituted with primers to viral (ISAV, IPNV and PMCV, SAV and PRV) and bacterial (*Tenacibaculum*) pathogens of key importance for Atlantic salmon aquaculture (Table 1). The gene composition of assay, sequences of primers, and functional annotations of genes are in Appendix A. For analyses, the individual RNA samples were adjusted at a final concentration of 10 ng/5 μL before reverse-transcription using the Reverse Transcription Master Mix (Fluidigm, South San Francisco, CA, USA). The resulting cDNA samples were added to the selected 96 primer pairs (100 µM) and the PreAmp master mix (Fluidigm, South San Francisco, CA, USA). Following 12 cycles of pre-amplification in the thermocycler TAdvanced (Biometra), the pre-amplified products were treated with the DNA-specific exonuclease I (New England BioLabs, Ipswich, MA, USA) and diluted in an SsoFast EvaGreen supermix with Low ROX (Bio-Rad, Hercules, CA, USA) and 20× DNA-binding dye sample loading reagent. These sample and primer mixes were transferred to twelve 48.48 dynamic array IFC chips, which were first primed in the BioMark IFC controller MX (Fluidigm, South San Francisco, CA, USA) for 60 min and then placed in the BioMark HD system (Fluidigm, South San Francisco, CA, USA). The qPCR cycling protocol contained a “hot start” activation step of 95 °C for 60 s, followed by 30 cycles of denaturation (96 °C, 5 s), annealing (60 °C, 20 s), and a melting step (60 to 95 °C, 1 °C/3 s). Fluidigm RealTime PCR analysis software v. 3.0.2 was used to retrieve raw qPCR results. Results were transferred in a relational database. The geometric means of two reference genes (*ef1f* and *rps20*), which showed stability across samples, were used for calculation of ΔCt, and ΔΔCt was calculated by subtracting the average ΔCt from each value. Differential expression of individual genes was assessed by criteria applied to microarray data: difference of ΔΔCt > |0.8| and *p* < 0.05. Gene expression in the gills was compared to a reference dataset (GRDS) with 595 healthy salmon covering different ages, seasons, areas of Norway, and aquaculture systems. Differences from the mean ΔΔCt values for each gene in GRDS were averaged in the samples. The deviation from GRDS for 2σ was set as the threshold value for outliers.

### 2.6. Bead Coupling and Multiplex Immunoassay

Analyses were performed as described previously [13]. The A-layer protein from *Aeromonas (A.) salmonicida subsp. salmonicida* ([18] produced as described in [19]) and whole-cell sonicate from *Moritella (M.) viscosa* type strain 1016/96 were included in the multiplex assay for the detection of antibodies induced with vaccination. DNP-keyhole limpet hemocyanin DNP-KLH (Calbiochem, Merck, Darmstadt, Germany) was used for the detection of nonspecific antibodies (NSAB). Antigens were coupled to distinct MagPlex^®^-C Microspheres (Luminex Corp. Austin, TX, USA) of different bead regions according to the manufacturer’s protocol using the Bio-Plex amine coupling kit (Bio-Rad, Hercules, CA, USA). DNP-KLH was used at 10 µg per 1× scale coupling reaction, and A-layer protein and *M. viscosa* sonicate at an amount of 12 µg and 7 µg, respectively. Briefly, beads were diluted in assay buffer (PBS with 0.5% BSA and 0.05% azide), and 5000 beads per region were added to each well. Beads were washed three times with assay buffer (30 s in the dark and on a shaker at 800 rpm), then kept for 120 s in a Bio-Plex handheld magnetic washer before the supernatant was poured off. Following initial titrations, plasma samples were diluted 1/4800 in assay buffer and added in duplicates on the plate, except for samples from parr (1st sampling) which were diluted 1/1200 in assay buffer. The plate was incubated for 30 min at RT in the dark and on a shaker at 800 rpm. All subsequent incubation and washing steps were performed similarly. Following incubation and washing, Anti Salmonid-Ig (H chain) monoclonal antibody (1:400, clone IPA5F12, Cedarlane, Burlington, ON, Canada) was added in each well. After incubation and washing, biotinylated goat Anti-Mouse IgG2a antibody (1:1000, Southern Biotechnology Association, Birmingham, AL, USA) was added in each well, and finally, after incubation and washing, Streptavidin-PE (1:50, Thermo Fisher Scientific, Waltham, MA, USA was applied. Plates were analyzed using a Bio-Plex 200 in combination with Bio-Plex Manager 6.1 software (Bio-Rad, Hercules, CA, USA). Each bead was classified by its signature fluorescent pattern and then analyzed for the median fluorescence intensity (MFI) of the reporter molecule.

### 2.7. ELISA

ELISA was used to measure total immunoglobulin (Ig) in plasma, as previously described [16]. All solutions were used at room temperature (RT). Nunc MaxiSorp plates (Thermo Fisher Scientific, Waltham, MA, USA) were coated with anti-salmonid Ig (heavy chain) monoclonal antibody supernatant (CLF004 from Cedarlane, Burlington, ON, Canada) diluted in carbonate buffer pH = 9.6 to an end concentration of 0.3 μg/mL and incubated for 48 h at 4 °C. Plates were washed 3 times with wash buffer (R&D systems) before the addition of blocking buffer (wash buffer + 4% horse serum (in-house) and an incubation for 2 h at RT). Following washing 3×, salmon plasma samples diluted 1/50,000 in sample diluent (wash buffer + 1% horse serum) were added to each well, and plates were incubated overnight at 4 °C. Plates were washed 4× before 100 μL primary antibody (rabbit-anti-salmonid Ig, CLF003AP from Cedarlane, ON, Canada) diluted 1:3500 were added to each well, and plates were incubated for 2 h at RT. Plates were washed 4× before 100 μL substrate (3, 3′, 5, 5′ Tetramethylbenzidine Liquid Substrate, TMB, T4444-100 mL from Sigma-Aldrich (Burlington, MA, USA)) was added to each well. Plates were wrapped in foil and incubated for 20 min at RT. Fifty microliters stop-solution (1M H2SO4) was added to each well before plates were gently shaken. Plates were read at 450 nm at Multiscan FC (Thermo Fisher Scientific, Waltham, MA, USA).

### 2.8. Statistical Analysis

Comparison of data within each cohort was analyzed with ANOVA followed with Tukey or LSD post-hoc test using Statistica 13.

## 3. Results

### 3.1. Fish Performance

One week before SWT, an average condition factor and smolt index of respectively 1.25 and 3.15 were recorded in fish from cohort 1 with an average weight of 87.4 g (+/−12.5 g). An average weekly site total mortality of 0.03% was recorded during the first 3 weeks after SWT, and the recorded feed conversion ratio (bFCR) was 1.0 during the same time period. After approximately 3 months in the sea, the average fish weight was 726 g, and the site total mortality and the bFCR were 0.04% and 1.0, respectively.

In cohort 2, an average condition factor of 1.27 and a smolt index of 3.80 were recorded the week before transfer in 113.85 g (+/−24.8 g) fish. An average weekly site total mortality of 0.02% and bFCR of 1.0 was recorded during the first 3 weeks after SWT. After approximately 3 months in the sea the average weight was 668.8 g, the site total mortality was 0.02% and the bFCR 1.0.

### 3.2. Gene Expression

#### 3.2.1. Pathogens

In addition to immune and stress genes, MGE assay included primers to selected viral and bacterial pathogens of major concern for Atlantic salmon aquaculture. Fish were negative for infectious salmon anemia (ISAV), infectious pancreatic necrosis (IPNV), and piscine myocarditis (PMCV) viruses in fin, gill and spleen at all sampling time-points. Piscine orthoreovirus (PRV) and Salmonid alphavirus (SAV) were detected in respectively 14 and 4 (sea only) fish from a total of 85 analyzed fish from both cohorts (Table 2). Tenacibaculum was found in 7 fish in freshwater and in 43 fish after SWT. Routine RT-PCR pathogen screening by Mowi detected IPNV in kidney from 6 of 10 fish analyzed from FW site 2 (sample analysis dated 14th of August 2018).

#### 3.2.2. Temporal Changes within Cohorts

Multigene expression assay was performed on three tissues: dorsal fin, gill, and spleen. Mean ΔΔCt is a simple metric reflecting overall expression levels of all immune genes included in the assay (Figure 2). The largest differences between the time-points within the cohorts as well as between the cohorts were observed in the dorsal fin (Figure 2A). The overall immune gene expression levels in the fin were generally lower in cohort 1 and showed no significant temporal changes. In comparison, fish from cohort 2 showed a high immune gene expression in parr, which was followed by a significant drop in smolts and a slow increase after SWT. Expression of selected VRG and AIS genes in the dorsal fin was consistent with this pattern; however, VRG showed a temporary peak in cohort 2 after 3 weeks in the sea (Figure 2D). In contrast to temporal fluctuations observed in the fin, the gill and spleen showed a steady increase in immune gene expression throughout the sampling period, although with minor fluctuations (Figure 2B,C). This increase in immune gene expression from parr to growers in the sea was more pronounced in cohort 1. Expression of VRG and AIS genes was highest at the last time-point (after 3 months in the sea) in the gill of fish from both cohorts (Figure 2E), whereas these genes showed highly variable expression in the spleen of fish from both cohorts (Figure 2F).

Comparison of immune gene expression in the gill with the gill reference data set (GRDS) [11] by ΔΔCt found deviation greater than 2σ (2SD) in only three out of 85 analyzed fish, all in cohort 2 and in parr, smolt, and grower after 3 weeks in the sea (Figure 3). The deviations from GRDS were normally distributed (Kolmogorov–Smirnov test).

### 3.3. Circulating Antibodies and IgM Repertoire

The kinetics of specific antibodies recognizing vaccine components and nonspecific antibodies binding to a model antigen were analyzed by multiplex immunoassay (Figure 4). In cohort 2, a significant increase in circulating antibodies was observed from parr to smolt, followed by a stabilization (*A. salmonicida* and NSAB) or further increase (*M. viscosa*) after 3 weeks in the sea and a steep decrease at the last time-point. Antibody levels in cohort 1 (FT) increased steadily throughout the sampling period but were at a lower level than in cohort 2 in smolts and after 3 weeks in the sea and until the last sampling point, when the antibody levels evened out between the two cohorts. The lower antibody levels in fish from cohort 1 were also reflected in the total plasma immunoglobulin levels (Figure 5). The correlation of antibody levels within cohorts was high, and it was higher between vaccine-specific and nonspecific antibodies (r = 0.83 and 0.75 for respectively A-layer and *M. viscosa*) than between the two vaccine-specific antibodies (r = 0.61) (not shown).

Sequencing of the IgM repertoire (Ig-seq) was performed to explore possible links between the kinetics of circulating antibodies and expansion of antibody-producing cells. Mounting of acquired immunity is associated with enhanced B-cell proliferation and IgM transcription within clones producing antibodies to recently delivered antigens, and both processes increase absolute (counts) and relative (frequencies) amounts of transcripts. Transcripts from newly expanded clones appear among the largest clonotypes, thus increasing the cumulative frequency (CF) of transcripts from the leaders. Here, CF of the fifty largest clonotypes (CF50) significantly increased in the gills of salmon from both cohorts and stabilized at high levels (Figure 6A,B). Similar kinetics was observed for the CF50 in the spleen in fish from cohort 2, whereas in cohort 1, CF returned to the initial level after 3 months in the sea (Figure 6C,D).

Unique immunoglobulin sequences or clonotypes mark B cells of clonal origin. The co-occurrence of identical clonotypes in the spleen and gill reflects the migration of B cells from the lymphoid (spleen) to the peripheral organ (gill). In Atlantic salmon, sharing of clonotypes between organs is relatively low under basal conditions but increases after strong immunization, followed by a gradual decrease because the largest clonotypes in the lymphoid organ are substituted with recently expanded clones. Recruitment of B cells to gills was markedly stimulated in smolts of both cohorts (Figure 7A,B), as illustrated by an increase in the number of clonotypes in the gill that were also detected in the spleen. In cohort 2, sharing steadily decreased afterward toward the end of the sampling period, while reduction was preceded with a peak after 3 weeks in the sea in cohort 1. At the last sampling time-point (3 months post-SWT), this parameter returned to the initial levels in both cohorts.

## 4. Discussion

Controlled experimental trials contribute essential knowledge on the immune system of Atlantic salmon; however, they do not necessarily reflect the complexity of immune responses in fish in commercial production. Until present, fish immunology has focused mainly on responses to pathogens, vaccination, stress, and other disturbances, while much less is known about the immune status of healthy fish in farming conditions. This descriptive case study is a step toward the development of methods for the assessment of the immune competence of farmed Atlantic salmon. The objective was twofold: (1) testing of newly developed tools on field material from commercial Atlantic salmon aquaculture and (2) assessment of the immune status of fish reared in different aquaculture systems.

The multigene expression assay (MGE) has been applied to material from controlled experiments and smolt batches with contrasting health conditions [11,16,20]. Here, we aimed to evaluate this tool in the field by assessing its ability to find temporal changes in the immune status of Atlantic salmon in relation to different aquaculture systems and production protocols and detect possible deviations from the norm. Multigene expression assay was performed on samples from the dorsal fin, gill, and spleen. Gill and dorsal fin are barrier tissues that allow rapid and easy noninvasive sampling, and spleen was selected as a lymphoid organ. Temporal changes in the gill and spleen were overall similar in both cohorts, showing a steady increase in the expression of immune genes throughout the sampling period, although with more fluctuations in the expression levels in cohort 2. Deviation from this trend was observed in the dorsal fin, where the expression level was highest in parr in cohort 2 (RAS). The reason for this is unknown, but it can be taken into account that the fin is constantly exposed to the environment, and that an RAS unit most likely has a more complex composition of the recirculating water. Overall, gene expression differences between salmon raised in cohort 1 (FT) and cohort 2 (RAS) were greatest at the first time-point in all analyzed tissues. Although fish originated from the same broodstock, immune genes in juvenile Atlantic salmon can be stimulated with various disturbances. For example, in a previous study using the MGE panel, we found higher immune gene expression followed by a decrease during smoltification in Atlantic salmon exposed to sustained developmental hypoxia [20].

Particular attention was paid to two diagnostic gene sets. An unexpected finding was a significantly higher abundance of VRG transcripts in the dorsal fin of cohort 2 (RAS) at first (parr) and third (3 weeks post-SWT) samplings. This was, however, not observed in the gills and spleen, and the detected increase in VRG in the fin was much smaller than what was previously observed in virus-infected Atlantic salmon [10], suggesting that this most likely does not indicate the presence of an unknown virus. Unspecific stimulation of VRG has been found in muscle injected with bacterial DNA–plasmid [21] and in leukocytes exposed to CpG [22].

External reference is essential for evaluation of the immune competence. Gene expression may fluctuate in a wide range without causing any visible problems. Compilation of reference data sets, including a gill reference data set (GRDS), is a crucial step in the development of tools for assessment of immune status of Atlantic salmon. Currently, to determine the boundaries of the norm, we can only rely on statistical parameters—a deviation of more than 2σ (*p* = 0.05) is not a pathology *per se* but can be considered as a warning sign. In the present study, with a normal distribution of the metric, the observed number of fish beyond the threshold of the GRDS was low, indicating a robustness of the external reference data set. No fish exceeded the 2σ threshold, and the mean expression of this set was 4.2 and 15.6 times lower than that of previously analyzed smolt batches with high mortality ([11] and unpublished data). Determination of biologically meaningful thresholds is of major importance for the future development of the MGE assay and will be updated as data on high-quality Atlantic salmon and fish with health problems accumulate.

Similar to the immune gene expression in the dorsal fin, the levels of both specific antibodies and NSAB were higher in cohort 2 (RAS) in smolts and in growers 3 weeks post-SWT; however, levels evened out between cohorts after three months in the sea. The difference in antibody levels in smolts between the cohorts is most likely explained by the difference in number of degree-days between the cohorts from vaccination to sampling of smolts, but it may also be influenced by the vaccines, the farm production protocol, or their combination. The persistently high levels of circulating NSAB in Atlantic salmon are in stark contrast to the almost complete lack of knowledge about their role. Moreover, strong correlation between levels of vaccine-specific and nonspecific antibodies is intriguing and is evidence in favor of diagnostic usefulness of NSAB, especially for multicomponent vaccines.

Vaccination may equally stimulate different populations of B cells, including those recognizing vaccine-delivered antigens and bystanders producing antibodies of broad reactivity. An alternative explanation is that the vaccine and model antigens, such as DNP-KLH, are recognized by the same antibodies. These antibodies may display a broad reactivity but most likely exhibit a lower affinity. This latter possibility is supported with markedly enhanced nonspecific binding of IgM in sera of carp (*Cyprinus carpio*) to a panel of human proteins after challenge with herpesvirus [23]. The authors recently observed the same in both vaccinated and SAV-infected Atlantic salmon, using a panel of poultry antigens. In that study, the ratio between total Ig and NSAB was calculated. Results showed that the ratio fluctuated over time, indicating that antibodies with a broad reactivity constitute a separate and variable proportion of the total amount of Ig (unpublished results).

Ig-seq was developed to assess the effects of vaccines and infections on the expansion of B-cell clones. The repertoire of IgM transcripts does not always reflect antibody responses, but in the present study, the two methods produced concordant results. Vaccine stimulated production of antibodies coincided with increased cumulative frequencies of the largest B-cell clonotypes (CF) and increased sharing of clonotypes between a peripheral organ (gill) and a lymphoid organ (spleen), which was highest in smolts (cohort 2) and after three months in the sea (cohort 1). We have reported enhanced traffic of B cells from the lymphatic organs to peripheral tissues and indicated its protective role in fish infected with salmonid alphavirus [15]. Following a peak in clonotype sharing, the traffic of B-cell clonotypes decreased and evened out between the cohorts after 3 months in the sea. To the best of our knowledge, transiently active migration in healthy fish observed in this study is a new finding. It remains to be determined whether recruitment of B cells to the barrier tissue was stimulated with vaccination, smoltification, or their combination.

## 5. Conclusions

In the present case study, tools developed for assessment of immune competence of Atlantic salmon were validated on field material from commercial aquaculture. Comparison of gene expression with an external gill reference data set suggested good immune status of fish reared in both FTS and RAS, as well as a robustness of the external reference data set. Differences between the cohorts that were observed before seawater transfer levelled out after several months in the sea. Future development of diagnostics requires defining the boundaries of normality as well as their biological significance.

## Figures and Tables

**Figure 1 genes-13-00736-f001:**
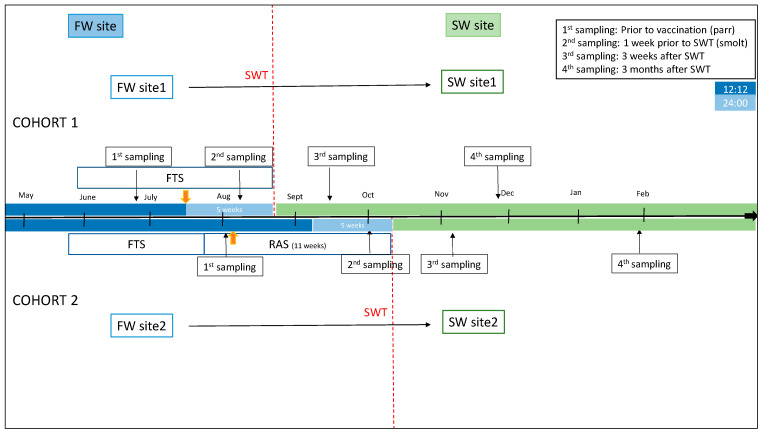
Production protocols and sampling regime. Cohort 1 consisted of FW site1 and SW site1. Cohort 2 consisted of FW site2 and SW site2. Time-points of samplings and SWT are indicated. Time-points for vaccination are indicated by yellow arrows. Light regimes in freshwater phase are indicated by light blue (24:00) and dark blue (12:12). Freshwater (FW), seawater (SW), seawater transfer (SWT), flowthrough system (FTS), recirculation aquaculture system (RAS).

**Figure 2 genes-13-00736-f002:**
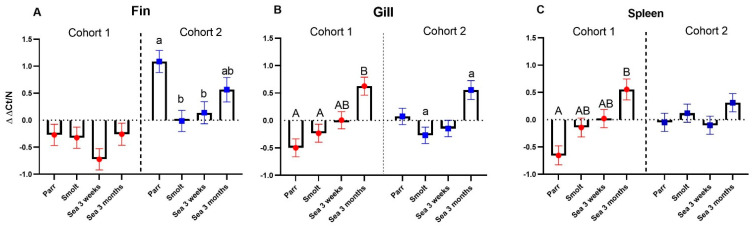
Temporal changes of gene expression in dorsal fin, gill, and spleen. Data are normalized within each cohort so that the mean of each gene in the entire data set is zero. (**A**–**C**): average expression of immune genes (mean ΔΔCt of all genes ± SE) in fin (**A**), gill (**B**), and spleen (**C**) (n = 9–12 individuals at each time-point). Bars within one cohort not sharing the same letters are significantly different (ANOVA, Tukey test, *p* < 0.05). The dotted lines separate cohorts. (**D**–**F**): Heatmaps present ΔΔCt for selected genes of innate antiviral immunity (VRG) and markers of acute inflammation and stress (AIS) in fin (**D**), gill (**E**), and spleen (**F**): VRG: ifit5 (interferon-induced protein with tetratricopeptide repeats 5), rtp2 (receptor-transporting protein 2), isg15 (interferon-stimulating gene 15), sacs (sacsin), irf1 (interferon regulatory factor 1), rsad2 (radical S-adenosyl methionine domain-containing protein 2, viperin) and ifn2a (interferon 2a). AIS: drtp1 (differentially regulated trout protein 1), mmp13 and 9 (matrix metalloproteinases), lect2 (leukocyte cell-derived chemotaxin 2), c7 (complement component c7), serpine1 (plasminogen activator inhibitor 1), arg2 (arginase-2), camp (cathelicidin), stbp (saxitoxin and tetrodotoxin-binding protein 2-like), saa5 (serum amyloid a-5) and il1b (interleukin-1 beta). Expression differences from the first time-point, parr (ΔΔCt > |0.8|, *p* < 0.05), are highlighted with underlined bold italics.

**Figure 3 genes-13-00736-f003:**
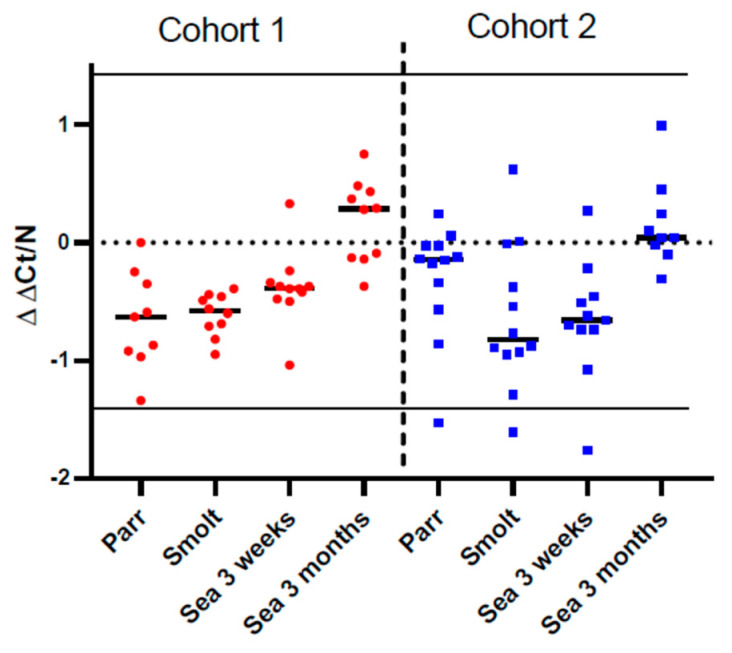
Mean deviations of immune gene expression in the gill from the gill reference data set (GRDS). Data are normalized so that the mean of each gene in the entire data set is zero (dotted line). Each mark corresponds to an individual (n = 9–12). Solid lines mark 2σ thresholds.

**Figure 4 genes-13-00736-f004:**
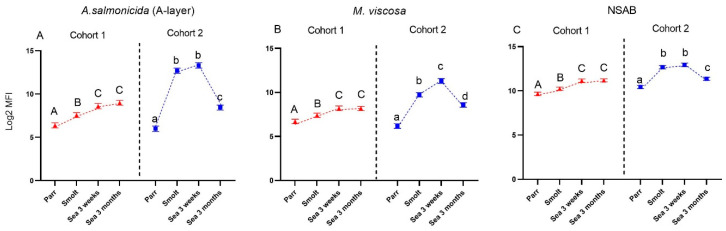
Multiplex immunoassays. Levels of circulating specific antibodies against A-layer of *A. salmonicida* (**A**) and *M. viscosa* whole-cell sonicate (**B**), and nonspecific antibodies (NSAB) (**C**). Data are displayed as mean log_2_MFI (median fluorescent intensity) + SEM (n = 12–15 individuals at each time-point). Data points within one cohort not sharing the same letters are significantly different (ANOVA, LSD test, *p* < 0.05).

**Figure 5 genes-13-00736-f005:**
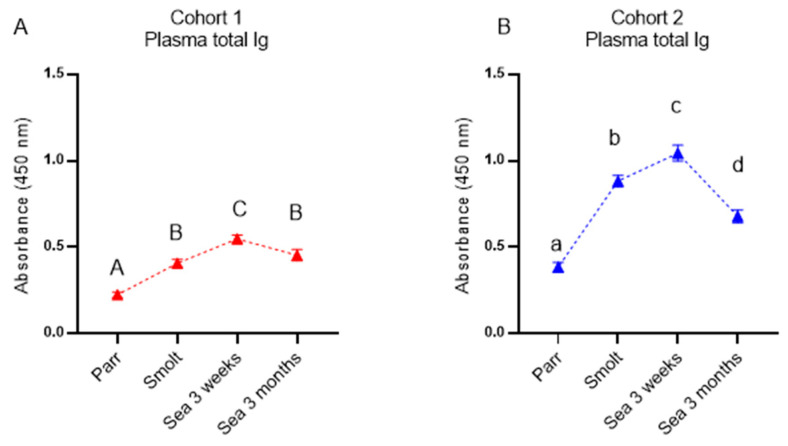
Total immunoglobulin (Ig) in plasma from Atlantic salmon from cohort 1 (**A**) and cohort 2 (**B**). Samples are analyzed by ELISA and at an absorbance of 450 nm. Data are presented as mean with SEM (n = 12–15 individuals at each time-point). Data points within one cohort not sharing the same letters are significantly different (ANOVA, LSD test, *p* < 0.05).

**Figure 6 genes-13-00736-f006:**
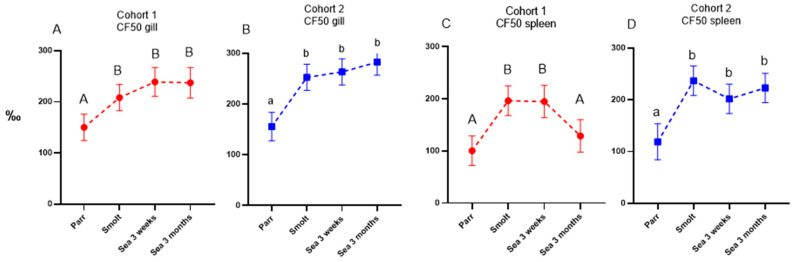
Sequencing of IgM variable region (Ig-seq). Cumulative frequencies (‰) of fifty largest clonotypes (CF50) in gill (**A**,**B**) and spleen (**C**,**D**). Data are presented as mean + SEM (n = 6–7). Data points within one cohort not sharing the same letters are significantly different (ANOVA, LSD test, *p* < 0.05).

**Figure 7 genes-13-00736-f007:**
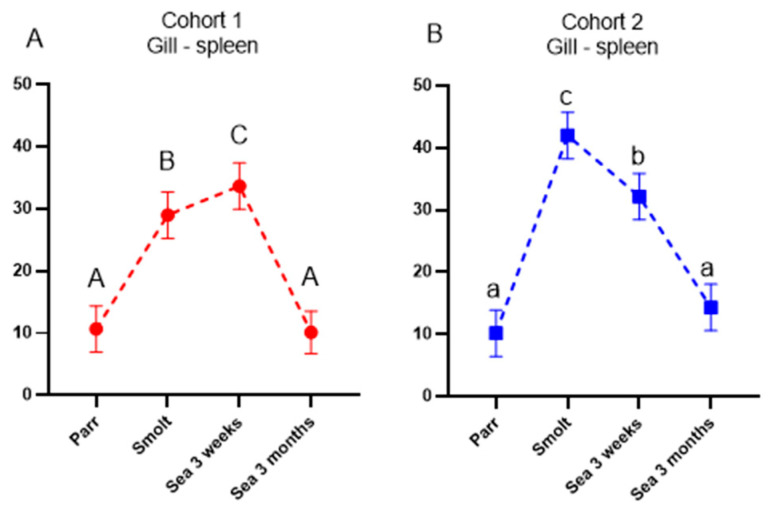
Co-occurrence of largest IgM clonotypes reflecting traffic of B cells from spleen to gill. (**A**), (**B**): the number of fifty largest gill clonotypes also detected in spleen. Data are presented as mean + SEM (n = 6–7). Data points within one cohort not sharing the same letters are significantly different (ANOVA, LSD test, *p* < 0.05).

**Table 1 genes-13-00736-t001:** Primers for viral and bacterial pathogens.

Pathogen	Abbreviation	NCBI Access/ PMID	Primer Sequence 5′→3′ (Sense, Antisense)	Fragment Length [bp]
Infectious salmon anemia virus	ISAV	PMID: 17058489	CAGGGTTGTATCCATGGTTGAAATG, GTCCAGCCCTAAGCTCAACTC	155
Infectious pancreatic necrosis virus	IPNV	NCBI: MH614932	CGACCGACATGAACAAAATCAGA, TCGTCGTTTCATCTGTCTTGCTA	182
Piscine myocarditis virus	PMCV	NCBI: HQ339954	AAGGAAAAGGGACGAGTATCTCA, AACCTCTTCTGTTGGTGATGTAAA	109
Salmonid alphavirus	SAV	NCBI: NC_003930	ATCTCACAGCTAACCCCTCCG, TAGCCAAGTGGGAGAAAGCTCT	165
Piscine orthoreovirus	PRV	NCBI: GU994022	TGCGTCCTGCGTATGGCACC, GGCTGGCATGCCCGAATAGCA	143
*Tenacibaculum*		NCBI: MF192916	TGCCTTTGATACTGGTTGACTTG, TTCGTCCCTCAGCGTCAGTATA	136

**Table 2 genes-13-00736-t002:** Numbers of fish with PCR detected pathogens in the multigene expression assays. Cohort, site, and sampling time-point are indicated.

Cohort	Sampling	PRV	SAV	*Tenacibaculum*	PRV and *Tenacibaculum*	SAV and *Tenacibaculum*
Cohort 1	FW 1	4	0	4	0	0
	SW 1	3	2	23	2	1
Cohort 2	FW 2	3	0	3	0	0
	SW 2	4	2	20	3	1

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
