# Peer review of "Evaluation of Immune Status in Two Cohorts of Atlantic Salmon Raised in Different Aquaculture Systems (Case Study)"

_genes, 2022, doi:10.3390/genes13050736_

Round 1

Reviewer 1 Report

In the present manuscript, the authors use their own RT-qPCR platform-MGE (ImCom)- to validate the platform, together with evaluate the immune performance of animals reared under FTS versus RAS facility. In addition, Ig-seq and immunoassays were realized. One of the main conclusions of the study is that there would not be a significant difference between animals reared in FST v/s RAS, after a time in SW. That is a relevant point from a practical view and could be useful for salmon farmers. The results/discussion seem to be almost descriptive, which can not reveal the significance and the deep mechanisms of immune response in the FTS/RAS context. 

Some general suggestions:

  1. please indicate if the fish, from MOVI, correspond to families with some genetic selection-management
  2. please indicate sex proportion in both cohort and pedigree structure.

3.       axis legends figure 2, unclear

  1. Include as supplementary information, raw data (CT) of RT-PCR analysis, the readers could welcome this information to replicate the results of the ms

Author Response

Responses to general suggestions: 

  1. please indicate if the fish, from MOVI, correspond to families with some genetic selection-management
  • The strain of fish used was Mowi. No specific disease traits were selected for and the only trait selected for was growth. This information has been added to the materials and methods.

  1. please indicate sex proportion in both cohort and pedigree structure.
  • The sex ratio in the cohorts was close to unity. Random sampling was performed and sex was not taken into account. This information has been added to the materials and methods.
  1. axis legends figure 2, unclear
    • The legend was edited.
  1. Include as supplementary information, raw data (CT) of RT-PCR analysis, the readers could welcome this information to replicate the results of the ms
  • Thank you for the comment. The Ct values are submitted as supplement.

Reviewer 2 Report

The manuscript by Lund et al. (Evaluation of Atlantic salmon Immunity) is a descriptive work providing an overview of newer tools for a somewhat more holistic view of immune status of cultured fish.  The developed tools are broad in scope and efficient in providing concise information for evaluation of population health. The comparison of NSAB to specific antibodies is particularly interesting given that each cohort was given different vaccinations. Currently the work appears to provide feedback for further evaluation of applicability. It would be expected that as these tools “mature” than can serve as a real proxy for understanding health of a population. I see only minor revisions needed which are listed below.

Some of the figures need to be revised at a higher resolution. Some fonts and lines appear a bit blurry.

There are some typographical errors and semantic oddities to correct:

Lines 47-48 – finish phrase “innate and adaptive arms of the immune response.”

Line 64 – unless I am mistaken I think it should read “response to pathogens” rather than “of”

Line 69 – complete the phrase “loss of balance” (balance of what?? Not equilibrium.)

Line 207 – correct RGDS to “GRDS”

Line 226-229  - This overly long sentence is grammatically incomplete and needs rewriting so as to make clear sense.

Lines 244-246 – This also is a sentence that is missing content and semantically lacks clarity. (“…beads were added Anti Salmonid-IgH…”???)

Table 2 - I do not see “sampling time point” indicated only site. Correct this table accordingly.

Figure 3 – Add to the caption the explanation for the dotted line. It appears in the text but it should also appear in the figure caption.

Line 375 – remove “the” from end of the line

Line 377 – “cells” should be singular

Figure 7 – caption states “hundred largest gill clonotypes”, but the previous text refers only to fifty largest clonotypes. Correct or explain this.

Line 496 – “on months” ??; correct this.

Line 499 – Add “the”, to read “in the sea”

Line 500 – Add “the”, to read as “To the best of our knowledge,…”

Line 510 – change “norm” to “normality” or “normal responses” etc….

Author Response

Responses to general suggestions: 

Some of the figures need to be revised at a higher resolution. Some fonts and lines appear a bit blurry.

  • Original figures will be submitted with revision.

There are some typographical errors and semantic oddities to correct:

Lines 47-48 – finish phrase “innate and adaptive arms of the immune response.”

  • Thank you for this comment, The phrase has been updated.

Line 64 – unless I am mistaken I think it should read “response to pathogens” rather than “of”

  • Thank you for noticing the error. This has been changed.

Line 69 – complete the phrase “loss of balance” (balance of what?? Not equilibrium.)

  • The phrase has been replaced by deviation from homeostasis.

Line 207 – correct RGDS to “GRDS”

  • Thank you for noticing this error. The abbreviation has been corrected.

Line 226-229  - This overly long sentence is grammatically incomplete and needs rewriting so as to make clear sense.

  • The sentence has been re-phrased.

Lines 244-246 – This also is a sentence that is missing content and semantically lacks clarity. (“…beads were added Anti Salmonid-IgH…”???)

  • Thank you for this comment, The sentence has been re-phrased for clarity.

Table 2 - I do not see “sampling time point” indicated only site. Correct this table accordingly.

  • Thank you for this comment, combination of cohort and sampling (FW or SW, 1 or 2) provides sufficient information.

Figure 3 – Add to the caption the explanation for the dotted line. It appears in the text but it should also appear in the figure caption.

  • Thank you for noticing this. This information has been added to the figure caption.

Line 375 – remove “the” from end of the line

  • This has been updated (now line 376).

Line 377 – “cells” should be singular

  •  

Figure 7 – caption states “hundred largest gill clonotypes”, but the previous text refers only to fifty largest clonotypes. Correct or explain this.

  • Thank you for noticing this error. This has been corrected.

Line 496 – “on months” ??; correct this.

  • This has been corrected.

Line 499 – Add “the”, to read “in the sea”

  • This has been added.

Line 500 – Add “the”, to read as “To the best of our knowledge,…”

  • Updated, thank you.

Line 510 – change “norm” to “normality” or “normal responses” etc….

  • Thank you for the suggestion. This has been changed.

Round 2

Reviewer 1 Report

The authors have incorporated the suggested